# Future intensification of compound and consecutive drought and heatwave risks in Europe

Samuel Jonson Sutanto[1], Confidence Duku[1,2], Merve Gülveren[3], Rutger Dankers[2], and Spyridon Paparrizos[1]

[1]Earth Systems and Global Change Group, Environmental Sciences Department, Wageningen University and Research, Droevendaalsesteeg 3a, 6708PB, Wageningen, the Netherlands
[2]Climate Resilience Group, Wageningen Environmental Research, Wageningen, the Netherlands
[3]Directorate of Climate Change, Ministry of Environment, Urbanization, and Climate Change, Ankara, Turkiye

**Correspondence:** Samuel Sutanto (samuel.sutanto@wur.nl)

**Abstract.** The risks of extreme weather events, such as droughts and heatwaves, are expected to rise across Europe due to global warming, leading to more severe and worsening impacts. These impacts become even more pronounced when compound and consecutive (CnC) drought and heatwave hazards occur. Yet, most studies on drought and heatwave have focused on single hazard events rather than CnC events and their potential impacts. This study aims to identify the future characteristics of both single and compound drought and heatwave hazards across Europe. More specifically, we analyzed changes in the total number of events, average duration, total duration, and frequency. Droughts were identified using the Standardized Soil Moisture Index (SMI) and heatwaves were detected using the Variable Threshold Method (VTM). Both hazards were assessed using bias corrected simulations from the Inter-Sectoral Impact Model Intercomparison Project (ISIMIP) models from 1953 to 2014 for historical period and from 2039 to 2100 for future climate scenarios under SSP1-2.6 and SSP5-8.5. Furthermore, we employ a machine learning (ML) approach to project the impacts of droughts and heatwaves, using Germany as a case study. The ML models were developed using hazard characteristics as predictors and drought and heatwave impact data as response variables. Results indicate that the number, duration, and frequency of both drought and heatwave events are projected to increase under SSP1-2.6, with even higher increase for SSP5-8.5, not only when analyzed independently but also as CnC hazards. This applies not only in the south but also across multiple other European regions. Drought hotspots were identified in the Western Europe, with projections showing an expansion toward the South and East under SSP1-2.6, and across nearly all of Europe under SSP5-8.5 except for the northern regions. Heatwave hotspots were primarily located in eastern and southern Europe, particularly in Russia, Italy, and Portugal. Future scenarios suggest that southern Europe will remain a key hotspot for heatwaves. The occurrence of compound drought and heatwave events was projected to increase sixfold compared to the reference period, while consecutive drought and heatwave events might rise by up to 3.5 times under SSP5-8.5. Additionally, results also reveal that drought impacts on economic, non-economic, and ecosystem sectors are projected to double in Germany, while heatwave impacts on human health and mortality may increase ninefold by 2100. Our findings highlight the need to consider CnC hazards and show once more the urgency of climate adaptation and mitigation in limiting impacts across multiple sectors.

## 1 Introduction

The occurrence of extreme weather events, such as droughts and heatwaves, has become more frequent and severe in recent decades due to climate change (Mukherjee and Mishra, 2021; Seneviratne et al., 2021; IPBES, 2021; EEA, 2024). The impacts of these events are intensified when droughts and heatwaves occur simultaneously (compound events) and/or consecutively (consecutive events) (Fischer et al., 2007; Van Lanen et al., 2016; Rakovec et al., 2022). For example, the major European droughts of 2003, 2010, 2018-2020, and 2022 were particularly severe and costly due to their combination with heatwaves and

other hazards such as wildfires (Robine et al., 2008; Ionita et al., 2017; Blauhut et al., 2022; Biella et al., 2024). These events heavily impacted sectors such as agriculture, energy, human health, and public water supply (Cammalleri et al., 2020; Biella et al., 2024). Recent drought events across Europe have resulted in estimated economic losses of approximately 9 billion EUR, with the highest losses recorded in Spain (1.5 billion EUR/year), followed closely by Italy (1.4 billion EUR/year) and France (1.2 billion EUR/year) (Cammalleri et al., 2020).

The impacts of drought and heatwave hazards, as single hazards and as compound and consecutive hazards (CnC), are expected to increase under global warming due to the increase of temperature and reduced precipitation in many land surfaces (Zscheischler et al., 2020; Seneviratne et al., 2021). For example, studies comparing the impacts of the 2003 and 2018 droughts and heatwaves on European ecosystems and forests concluded that the 2018 event was more extreme than that of 2003 (Buras et al., 2020; Schuldt et al., 2020). As a result, the negative impacts of the 2018 hot drought on ecosystems were not only

stronger but also affected a larger area. Additionally, Cammalleri et al. (2020) projected that agriculture and energy sectors will experience the highest impact compared to others, while Naumann et al. (2021) estimated that economic losses due to drought in Europe could rise to 65 billion EUR by 2100. The severity of these impacts, especially when combined with other hazards, call the urgent need to manage and minimize the impacts associated with future CnC drought and heatwave events as well as to develop effective adaptation strategies.

Several studies have analyzed the occurrence of droughts and heatwaves in Europe under global warming, but relatively few have discussed the characteristics of CnC drought and heatwave events, such as duration and frequency (Samaniego et al., 2018; Spinoni et al., 2018; Tripathy et al., 2023; Tripathy and Mishra, 2023). In addition, these studies primarily focused on either single or compound events, neglecting consecutive events. Importantly, projections of the impacts of droughts and heatwaves in various sectors, beyond just the hazards themselves, are generally lacking. Naumann et al. (2021) study is pioneering in

analyzing future drought impacts in terms of economic losses by integrating damage functions with hazard and exposure components.

   Another promising approach for impact prediction is by applying machine learning (ML) algorithms. Previous studies have utilized ML approaches to link reported drought impacts with hazards and to forecast drought impacts (Stagge et al., 2015; Bachmair et al., 2017; Sutanto et al., 2019a). Stagge et al. (2015) and Bachmair et al. (2017) employed logistic regression,

hurdle models, and random forest algorithms to develop drought impact models using historical meteorological drought indices

as predictors and impact records from the European Drought Impact Inventory (EDII, Stahl et al. (2016)) as the response variable in a binary setting. Building on this, Sutanto et al. (2019a) developed seasonal drought impact forecasting models using logistic regression and random forest techniques based on forecasted hydrometeorological drought indices and EDII database. The outcome of the ML models is likelihood of impact occurrences (LIO). The EDII was also used to validate the simulated impacts. Despite these advances, the application of ML to project future drought and heatwave impacts, as opposed to hazards, has so far been limited.

In this study, we analyze the characteristics of future hydrological drought represented by soil moisture drought and heatwave events including their CnC events across Europe. Furthermore, we also aim to predict the impacts of droughts and heatwaves under both low and high Shared Socioeconomic Pathways scenarios (SSP1-2.6 and 5-8.5, respectively). In our study, we define compound drought and heatwave (CDH) hazards if these events occur simultaneously on the same day and same location (Leonard et al., 2014; Liu and Huang, 2015) while consecutive drought and heatwave (CoDH) is defined if these hazards occur successively or cumulatively over time without being interrupted by a zero-hazard day (Sutanto et al., 2019b; Vitolo et al., 2019).

The paper is structured as follows: Section 2 describes the methodology used in this study, Section 3 presents the results, Section 4 discusses the findings, and finally, Section 5 concludes the study. Section 2 is further divided into six sub sections, describing the hydrometeorological data used in the study (Section 2.1), the drought and heatwave impact database (Section 2.2), methods for identifying droughts and heatwaves (Section 2.3), identification of CnC events (Section 2.4), drought and heatwave characteristics (Section 2.5), and impact projections using machine learning (Section 2.6). In Section 3, we present the characteristics of single hazards in Section 3.1, CnC hazard characteristics in Section 3.2, a regional summary of both single and CnC hazards across Europe in Section 3.3, and projections of drought and heatwave impacts in Section 3.4.

## 2 Methods

### 2.1 Hydrometeorological data

Soil moisture data were obtained from the ERA5-Land and the Inter-Sectoral Impact Model Intercomparison Project (ISIMIP) (Warszawski et al., 2013). ERA5-Land monthly soil moisture data were downloaded from the Copernicus Data Store (CDS), while in ISIMIP, soil moisture data (ISIMIP3b) were generated using the CWatM hydrological model (Burek et al., 2020), driven by five Coupled Model Intercomparison Project Phase 6 (CMIP6) global climate models: GFDL-ESM4, IPSL-CM6A-LR, MPI-ESM1-2-HR, MRI-ESM2-0, and UKESM1-0-LL. These ISIMIP models were run for both historical and future scenarios under SSP1-2.6 and SSP5-8.5 pathways. ISIMIP model data, originally at a spatial resolution of 0.5° x 0.5°, were resampled to 0.1° x 0.1° using bilinear interpolation approach to match with ERA5-Land resolution. The resampled was done to facilitate an easy bias-correction of the ISIMIP soil moisture data with ERA5-Land data, and should not be regarded as an attempt to downscale the ISIMIP data. The ISIMIP soil moisture data were then bias corrected using the delta method, with ERA5-Land as a benchmark (Hawkins et al., 2013).

ISIMIP hourly near-surface temperature data are available for five CMIP6 global climate models. These hourly temperature data were then converted to daily maximum and minimum temperature for heatwave analysis. The ISIMIP temperature data have already been bias corrected and therefore, we did not perform bias correction analysis (Lange and Büchner, 2021). Same as soil moisture data, a bilinear interpolation approach was applied to resample the temperature data to a 0.1° x 0.1° spatial resolution to ensure consistency with ERA5 Land.

The resampling approach from coarse to high resolution data was employed in this study because high resolution results are needed to develop impact prediction algorithms using ML at country scale, here is Germany. Using a coarse resolution for impact prediction will result in limited number of grid cells. We did not apply statistical or dynamical downscaling techniques, and as such the resampling of the ISIMIP data did not substantially change the climate change signal that is contained in these data.

The uncertainties within the ISIMIP models have been investigated in previous studies (Samaniego et al., 2017; Pechlivanidis et al., 2017; Vetter et al., 2017; Hattermann et al., 2018). These studies concluded that most of the variability stems from the climate models rather than the hydrological models. Furthermore, uncertainties tend to be higher in dry basins than in wet basins (Samaniego et al., 2017; Pechlivanidis et al., 2017). Nevertheless, ISIMIP model data has been utilized to study the extreme events, such as droughts, floods, and heatwaves (Samaniego et al., 2017; Pechlivanidis et al., 2017; Tabari et al., 2021; Messori et al., 2025), suggesting its robustness for such applications. In this study, we also tested drought and heatwave analysis derived from ISIMIP models and ERA5 Land. The results show that the simulated number of drought events from ISIMIP models aligns closely with ERA5 Land, with a median difference of only 7% (Supplementary Figure S1a). For heatwaves, ISIMIP models slightly underestimate their frequency compared to ERA5 Land, with 75 percentile of events reaching 75 in ERA5 land and 59 in ISIMIP models (Supplementary Figure S1b). These findings support previous studies, which report higher uncertainty from the climate models than the hydrological model.

Heatwave and drought analyses were performed individually for each model. Changes in dry hazard characteristics as single and compound events are determined by calculating the difference between future and historical events (future minus reference period). For drought, we used historical data from January 1953 to December 2014 (12 months x 62 years) as reference period and from January 2039 to December 2100 for future period (62 years). Same period was chosen for heatwave analysis but we only considered extended summer period from May to October (184 days x 62 years). Detailed data used in the study is presented in the Supplementary Table S1.

## 2.2 Drought and heatwave impact databases

Drought and heatwave impact databases were compiled from the European Drought Impact Inventory (EDII) (Stahl et al., 2016), the international disaster database (EM-DAT) (Jonkman, 2005), and data mined from scientific and grey literature in English. We followed the format of EDII to compile the heatwave and used the same impact categories. Drought impact data was then grouped into three different sectors, which are economic, non-economic, and ecosystem sectors (Biella et al., 2024). The economic sector consists of agriculture and livestock, forestry, aquaculture and fisheries, energy and industry, waterborne transportation, and tourism and recreation impacts. The non-economic sector includes public water supply, water quality, air

quality, health and public safety, and water access conflicts. The ecosystem sector covers impacts on freshwater ecosystems, terrestrial ecosystems, soil systems, and wildfires. This categorization was based on the drought impact divisions established by the Dita network, IAHS group (Biella et al., 2024).

Heatwave impacts reported from data mining include agriculture, air quality, wildfire, and economic in addition to the excess mortality statistics often reported. The collected impact data other than excess mortality, however, are too limited and can be combined to human impacts. For instance, wildfires triggered by heatwaves are often also reported having an impact of mortality. Thus, we decided to combine this into human impact. Due to limitations in the reported drought and heatwave impact data, this study only focuses on impact analysis for Germany.

## 130   2.3    Drought and heatwave indexes

Droughts in soil moisture were identified using the Standardized Soil Moisture Index (SMI) following the construction of the Standardized precipitation Index (SPI) (McKee et al., 1993). The SMI provides a measure of soil moisture dryness by quantifying the deviation from the long-term mean, i.e. number of standard deviations. The SMI was calculated by fitting a probabilistic distribution on monthly soil moisture data. To compute the SMI, the monthly historical soil moisture data was 135   transformed into 12 distributions, corresponding to the month of the year. Gamma distribution was employed in this study and is described by two parameters: $\alpha$ (the shape parameter) and $\beta$ (the inverse scale parameter). The gamma distribution has quite a flexible shape parameter, which is suitable for a wide range of drought application in EU (Stagge et al., 2015; Sutanto and Van Lanen, 2021). These distribution parameters then were used to calculate observed and future drought events (for further details, refer to Sutanto et al. (2020a)). The use of distribution parameters derived from the historical period to estimate future 140   droughts implies that no adaptation to climate change is assumed.

     Heatwave events were identified using the daily variable threshold method (VTM) derived from the historical data (Sutanto and Van Lanen, 2021). At each grid point, the threshold of 90th percentile of the daily maximum and minimum temperature for the historical period was calculated and nine days centered moving window approach was applied to consider the temporal variation of temperature. We identified a heatwave event if both maximum and minimum temperatures exceed the thresholds 145   for at least 3 days during the extended summer period, from May to October (Lavaysse et al., 2018; Sutanto et al., 2019b). Two successive heatwave events are considered independent if they are separated by a minimum of four days (temperature below threshold), otherwise, it is taken as one single event (Mukherjee and Mishra, 2021). Similar to drought, the threshold levels derived from historical data were applied to analyze future heatwaves, meaning that no adaptation to climate change is assumed.

## 150   2.4    Compound and consecutive (CnC) events

We identified compound and consecutive event as drought and heatwave occurred at the same time and place (concurrent) and one after another at the same time and place (sequential), respectively (Leonard et al., 2014; Liu and Huang, 2015; Vitolo et al., 2019; Sutanto et al., 2019b). Therefore, the definition of consecutive event differs from compound event. If drought occurs

**Table 1.** Examples of drought and heatwave combinations in compound and cascading events. CDH stands for compound drought and heatwave and CoDH stands for cascading drought and heatwave.

| No | Example of hazard event | Event name |
|----|------------------------|------------|
| 1 | 0,3,3,3,0 | CDH |
| 2 | 0,1,2,2,0 | CoDH |
| 3 | 0,2,2,1,0 | CoDH |
| 4 | 0,2,3,1,0 | CoDH |
| 5 | 0,1,3,2,0 | CoDH |

after heatwave event is over (here the temperature back to normal-high, not extreme), then we define this event as consecutive
and not compound/concurrent because there is only one single hazard left in the end.

To analyse the CnC events, binary maps consist of the number 1 for heatwave and 2 for drought were generated if the month is identified as drought or heatwave month. For no hazard month, 0 value is applied. The next step was overlapping the monthly maps of individual hazards by aggregate the grid cell values. Compound event is identified if the summed value is 3 (1+2), meaning that heatwave and drought occurred at the same time and grid cell. A consecutive event occurs if two different events
happen sequentially without being interrupted by a zero-hazard month. The consecutive event can consist of two single hazards (e.g., drought and heatwave or vice versa), or single and compound events (e.g., drought and CDH or heatwave and CDH). For instance, the binary time series of the events like "0-1-2-2-0", "0-2-2-1-0", "0-2-3-1-0", and "0-1-3-0" indicate consecutive events of heatwaves-drought (1, 2), drought-heatwaves (2, 1), drought-compound drought heatwaves-hetawaves (2, 3, 1), and heatwaves-compound drought heatwaves (1, 3), respectively. This means, CDH map contains values 0 and 3 while CoDH map
contains values above 0 for more than 1 (see Table 1). Previous studies by Sutanto et al. (2019b) and Vitolo et al. (2019) provide detailed descriptions of the CnC events. Supplementary Figure S2 shows examples of CDH events from May to October 2018 and from June to August 2019, and CoDH events from May to November 2018 and from June to October 2019 in Germany. There were two drought events occurred in February 2018 and April 2019.

### 2.5 Extreme event characteristics

We defined the characteristics of drought and heatwave hazards both as single and CnC as the total number of events, average duration of events, total duration of events, and frequency of events. The total number of events is defined as total hazard events in a particular period, either the reference or future period, summed across the 62 years. The total duration of hazard events is defined as the total duration of hazards during the analyzed period summed across the 62 years, measured in months for droughts, and days for heatwaves. The average duration of hazard events is calculated by dividing the total number of events
with the total duration of events. Lastly, the frequency of events is defined as the average number of hazard events in a year. For CnC analysis, we assigned the heatwave binary month if there was at least one heatwave event in that particular month (see previous sub-section). Supplementary Table S2 provides detailed information of dry hazard characteristics.

## 2.6 Hazard projections using machine learning

Drought index derived from the SMI, and the updated drought impact data for Germany were utilized to develop drought impact prediction models using an Extreme Gradient Boosting (XGBoost) ML approach (Friedman, 2001; Mardian et al., 2023). The ML models to predict drought impacts on economic sector, non-economic sector, and ecosystem sector were trained using soil moisture data, SMI, and drought duration as the predictors and drought impact data as the response variable. The reported drought impact data were categorized into economic sector, non-economic sector and ecosystem sector (see previous sub-section). However, for heatwave impacts, for which there was a limited data record, the ML model was developed using reported impact data on human mortality as the response variable and heatwave intensity, duration, and number of events as the predictors. The drought and heatwave impact data were initially pre-processed by ensuring coverage for all months of the year, filling in missing months with zero values to indicate the absence of extreme events. This was achieved by creating a multi-index using the 'year' and 'month' columns and expanding the data to span all possible combinations. The resulting dataset was merged with the original data to ensure a comprehensive timeseries, with missing values filled appropriately.

For both drought and heatwave reported data, the Synthetic Minority Over-sampling Technique (SMOTE) (Chawla et al., 2002; Lemaître et al., 2017) was used to address inherent class imbalances that could potentially lead to biased models. Class imbalance poses a significant challenge to many ML algorithms, particularly those based on classification trees or other forms of supervised learning. When the target variable's distribution is highly skewed, the model tends to be biased towards predicting the majority class, as it can achieve a superficially high accuracy simply by ignoring the minority class. Droughts and heatwaves are essentially rare events occurring infrequently as reported in the datasets. This imbalance can cause the model to overlook or underpredict the occurrence of such extreme events. By applying SMOTE, which effectively mitigates the class imbalance by generating synthetic samples for the minority class, thereby performing an "up-sampling" procedure, we generated synthetic samples for the minority class, thus enabling the model to better capture the underlying patterns of these infrequent but impactful events. This approach enhances the model's ability to predict the likelihood and impacts of droughts and heatwaves, leading to more reliable and robust predictions.

To model the relationships between predictors and responses, separate XGBoost models were trained for each of the three categorized drought impacts, as well as for heat impact. Hyperparameter tuning was performed to optimize model performance, with parameters such as learning rate, gamma, and maximum depth iteratively adjusted to identify the optimal configuration for predictive accuracy. Data splitting into training and testing sets ensured a robust evaluation and validation of the models. The performance of the models in predicting drought impacts for various sectors, as well as heatwave impacts on humans, was evaluated using the Relative Operating Characteristic (ROC) score (Mason, 1982).

# 3 Result

## 3.1 European drought and heatwave characteristics

Based on the median ensemble of ISIMIP model simulations (Warszawski et al., 2013) from 2039 to 2100 (62 years), the occurrence of drought events in Europe is projected to increase by at least 40 events under the SSP5-8.5 scenario compared to the reference period of 1953-2014 ( 84 events, Fig. 1a). Countries such as Spain, France, the Netherlands, Belgium, Italy, the Balkan regions, and Finland will experience a total drought month (duration) exceeding 220 months over 62 years (744 months), nearly twice as long as the reference period (Fig. 1b) (Samaniego et al., 2018). On average, drought duration will increase by up to half a month in these regions (Fig. 1c). Similarly, the frequency of droughts across Europe is expected to increase by 1 event more than in the past (Fig. 1d) (Spinoni et al., 2018). Under the SSP1.2-6 scenario, the increase in drought characteristic across Europe is approximately half of that under SSP5-8.5 and even lower in western Russia for average drought duration (Supplementary Fig. S3). For instance, drought events are projected to increase by around 20 events in Germany under SSP1-2.6 and around 40 events under SSP5-8.5.

The increase in the number of drought events and frequency (number of events per year) in eastern Europe is clearly visible when comparing the number of drought events during the reference period to the far future under SSP5-8.5 scenario (Fig. 1a and 1d). For the drought hotspot, however, Western Europe was identified as a drought hotspot ( 90 events), with fewer drought events occurring in eastern Europe (Supplementary Fig. S4a). Drought hotspot regions are projected to spread towards the south and east under SSP1.2-6, experiencing approximately 20-30 more drought events than in the past (Supplementary Fig. S4b). Under the SSP5.8-5 scenario, almost all of Europe, except for northern Norway and Sweden, is expected to become drought hotspot regions, with at least 40 more events than in the past, representing an increase of over 44% (Supplementary Fig. S4c).

The increase of European temperature due to climate change strongly influences heatwave characteristics (Russo et al., 2015). The number of heatwave events exhibits substantial changes in many regions, especially in southern Europe, where more than 350 events in 62 years (>5 events per year) are anticipated under SSP5-8.5 (Fig 2a). Sweden and Ireland are expected to experience 150 more heatwaves than during the reference period. The changes in total and average heatwave durations are particularly pronounced in southern Europe (Amengual et al., 2014; Molina et al., 2020). In this region, more than 4000 days (36%) between 2039 and 2100 (total 11,408 days) are categorized as heatwave days compared to the past period (1953-2014, Fig. 2b). This means that southern Europe will experience heatwaves for around two months each year during the extended summer period from May to October. Future heatwave events will last at least six days longer in Spain, Italy, and the Balkan region (Fig. 2c). On average, two more heatwave events will occur in the future in southern, western, and parts of eastern Europe (Fig. 2d). The changes in characteristics of future heatwave events will be ∼50% lower under SSP1-2.6 than SSP 5-8.5 (Supplementary Fig. S5). For instance, Germany may experience 150 more heatwave events than the reference period, with a total heatwave duration of 1000 days under SSP1-2.6.

Unlike droughts, which often occurred in western Europe and the UK during the reference period (Supplementary Fig. S4a), the hotspot regions for heatwave events are identified in eastern Europe, mainly in Russia, Italy, and Portugal (Supplementary

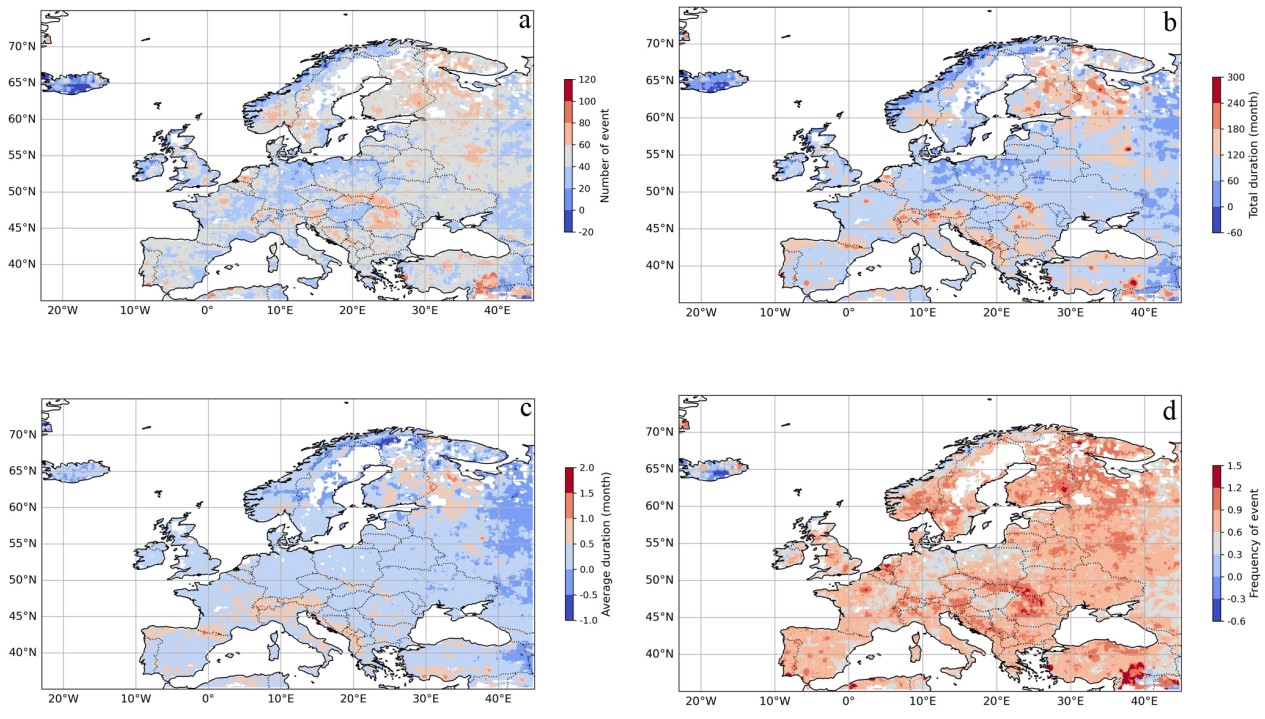

**Figure 1.** Changes (future-reference periods) in drought characteristics across Europe under SSP5-8.5 based on the median ensemble of the ISIMIP models. a) Changes in the number of drought events, b) changes in total drought duration in month, c) changes in average drought duration in month, and d) changes in frequency.

Fig. S6a). Global warming, however, shifts these hotspot regions into the Mediterranean, extending from Spain, Italy, and the Balkan countries to southeastern Europe. The number of heatwave events in these regions is projected to be approximately 300 and 500 events higher than the reference period under SSP1-2.6 and SSP5-8.5 scenarios, respectively (Supplementary Fig S6b,c). The rate of increase in heatwave events is less pronounced in the UK, Scandinavian countries, and northeastern Europe.

## 3.2 The characteristics of compound and consecutive events

The changes in compound drought and heatwave (CDH) characteristics across Europe in a warming world under SSP5-8.5 are presented in Figure 3. Overall, the number of CDH events increases by 15 and 30 events more than the reference period under SSP1-2.6 and SSP5-8.5, respectively (Supplementary Fig. S7a and Fig. 3a). Almost all European regions, except Sweden and northern UK, will experience similar changes under extreme climate change scenarios. However, some countries, such as France, western Germany, Switzerland, Austria, northeastern Spain, Norway, and Moldova will experience significant changes even under SSP1-2.6 (Supplementary Fig. S7a). The highest changes in CDH duration are found in western, southern, and

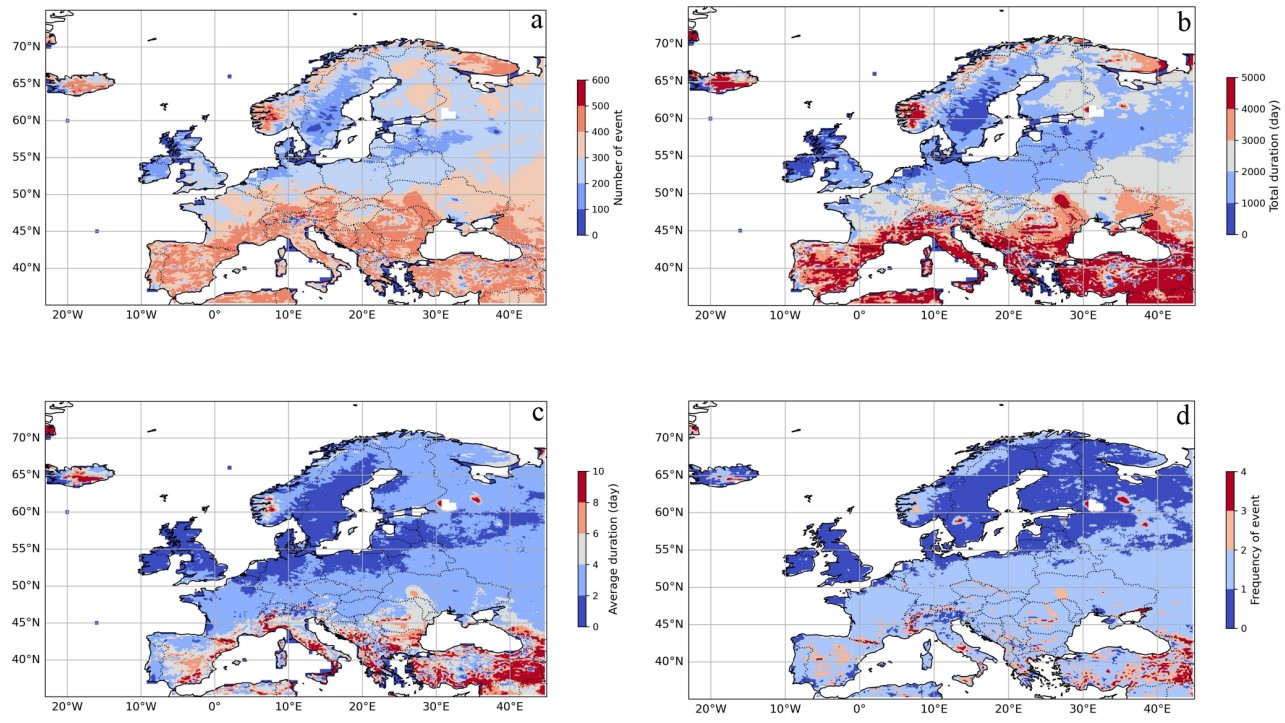

**Figure 2.** Changes (future-reference periods) in heatwave characteristics across Europe under SSP5-8.5 based on the median ensemble of the ISIMIP models. a) Changes in the number of heatwave events, b) changes in total heatwave duration in day (total number of day for 62 years is 11,408 days), c) changes in average heatwave duration in day, and d) changes in frequency.

eastern Europe (Fig. 3b,c), with a pattern similar to changes in total heatwave duration. This indicates that changes in CDH duration are driven by heatwave duration. The frequency of CDH events will increase on average by around 0.4 events per year under SSP5-8.5, with a maximum frequency of 0.76 events per year (Fig. 3d).

The changes in consecutive drought and heatwave (CoDH) characteristics under extreme warming are similar to CDH (Supplementary Fig. S8). However, CoDH characteristics under SSP1-2.6 show higher changes than CDH (see Supplementary Fig. S7 and S9). On average, the occurrence of CoDH events under SSP1-2.6 is 19 events more than the reference period (compared to 15 under SSP5-8.5). Many regions in Europe will experience higher number of CoDH events than CDH under SSP1-2.6, leading to a higher frequency of CoDH events. In terms of duration, the changes are more pronounced for total

CoDH than total CDH, with total durations being 77 and 55 months longer than the reference period, respectively (Sutanto et al., 2019b). In contrast, the average duration of CaDH events (0.8 month) is shorter than that of CDH events (1.1 month). Our results indicate that under SSP1-2.6, CoDH events will occur more frequently across Europe but with shorter durations compared to CDH events.

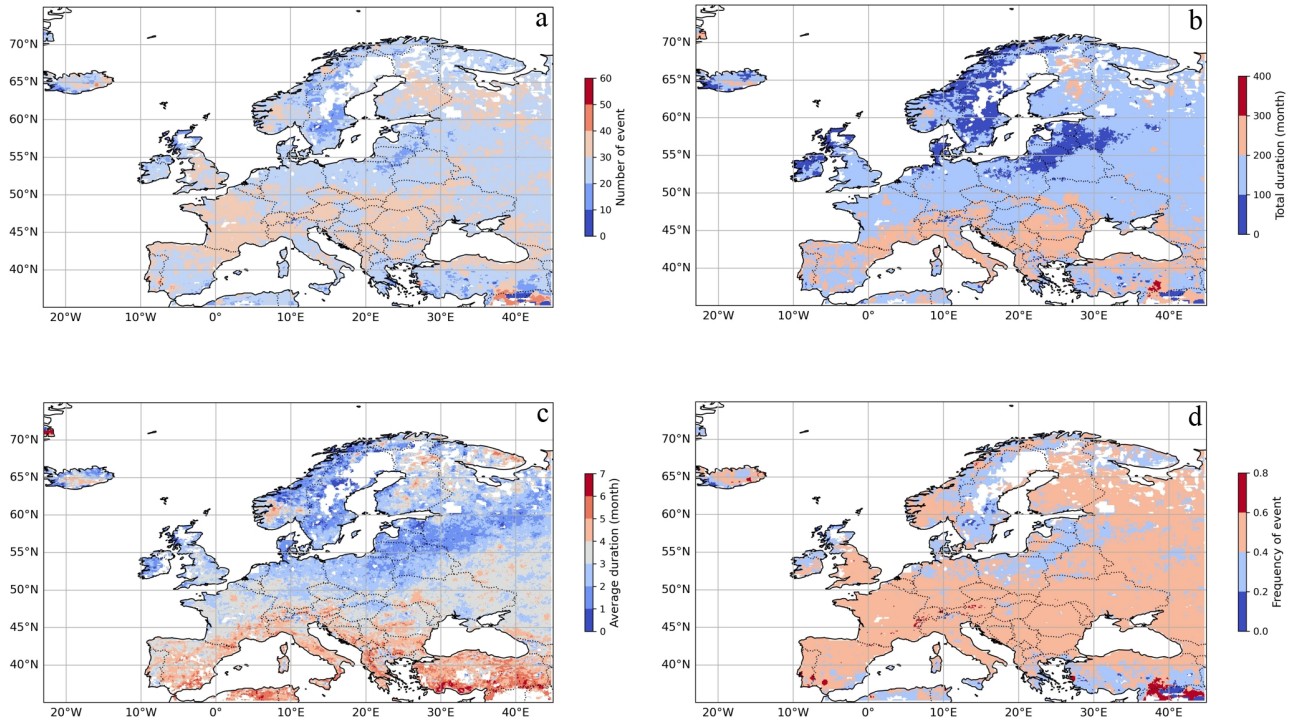

**Figure 3.** Changes (future-reference periods) in compound event characteristics across Europe under SSP5-8.5 based on the median ensemble of the ISIMIP models. a) Changes in the number of compound events, b) changes in total compound duration in month, c) changes in average compound duration in month, and d) changes in compound frequency.

Analyses of individual CnC events from the reference period reveal that northern Portugal, Italy, and eastern Europe used to be hotspot regions for both compound and consecutive events (Fig. 4a,d) (see also Niggli et al. (2022)). Under the SSP1-2.6 scenario, some parts of France, the Alps, southern Germany, and Moldova will become new hotspot regions for CnC dry hazards, associated with a higher increase in CnC events in these regions compared to others (Fig. 4b,e). For example, in eastern Europe, the number of CDH events is projected to increase from 10 events in the past to 25 events in the future under SSP1-2.6, more than doubling. In the Alpine region, the number of CDH events is expected to increase sharply from 4 events in the past to 35 events in the future, which is almost ninefold. A significant difference between future CDH and CoDH hotspot regions is observed as the climate becomes more extreme. While some regions in northern Europe may not emerge as hotspots for CDH, most European regions will become hotspot regions for CoDH events (Fig. 4c,f).

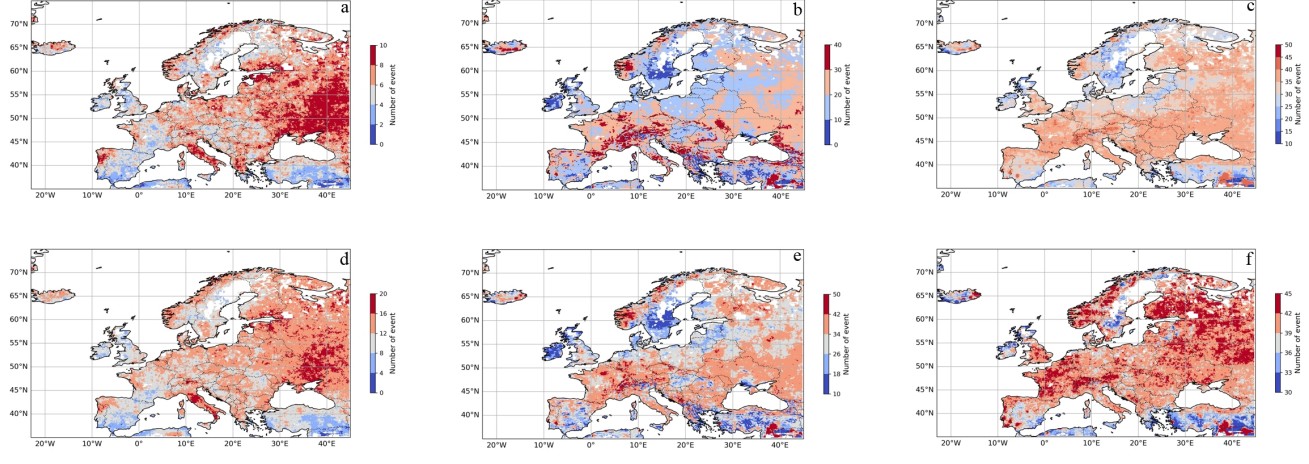

**Figure 4.** Hotspot regions for compound drought and heatwave (CDH) derived from the number of CDH event across Europe. a) Hotspot regions based on historical period (1953-2014), b) hotspot regions based on future period under SSP1-2.6 (2039-2100), and c) hotspot regions based on future period under SSP5-8.5 (2039-2100). d,e,f) Same as a, b, and c but for consecutive drought and heatwave (CoDH). The number of CnC events was obtained from the median ensemble of the ISIMIP models.

## 3.3 Summary of single and CnC dry hazards in Europe

Figure 5 summarizes the characteristics of droughts, heatwaves, and CnC events, averaged from all grid cells in each European
region (Supplementary Fig. S10) from past to future scenarios. Western Europe (WE) has historically experienced a high number of drought events (88.9 events), whereas Eastern Europe (EE) has had slightly fewer (77.7 events). However, EE has experienced the longest drought durations, with an average duration 0.2 to 0.3 months longer than that of other regions (Sutanto and Van Lanen, 2020). These findings align with Samaniego et al. (2018), who projected EE to experience the longest drought duration under 3°C warming scenario. Future projections under SSP5-8.5 indicate that WE will continue to experience the
highest number of drought events, with an average of 130.7 events. This increase is likely influenced by the high number of events in France, identified as a drought hotspot in WE (Spinoni et al., 2018). Nonetheless, the increase in drought events in WE (41.8 events) is lower compared to EE, where the rate of increase is projected to be 46.2 events. Conversely, Southern Europe (SE) is projected to face the longest drought durations (Samaniego et al., 2018; Sutanto et al., 2019b), with an average total duration extending up to 231.8 months, doubling from historical values. Both WE and SE are expected to see an increase
in the frequency of drought events (2.1 events per year) (Forzieri et al., 2014).

Eastern Europe (EE) experienced the highest number of heatwave events (65.9 events) in the historical period, followed closely by SE with 65.4 events, WE with 54.4 events, and Northern Europe (NE) with 51.9 events (Figure 5). Regarding heatwave duration, SE reported the longest total duration, averaging 303.5 days, with EE closely following at 299.4 days. The average duration of heatwaves was 4.6 days in SE and 4.5 days in EE. Future projections indicate that SE is expected

| | Number of events | | | Total duration | | | Average duration | | | Frequency of events | | |
|---|---|---|---|---|---|---|---|---|---|---|---|---|
| | Historical | SSP1-2.6 | SSP5-8.5 | Historical | SSP1-2.6 | SSP5-8.5 | Historical | SSP1-2.6 | SSP5-8.5 | Historical | SSP1-2.6 | SSP5-8.5 |
| **Drought** | | | | | | | | | | | | |
| NE | 83.9 | 106.3 | 123.7 | 118.8 | 157.6 | 204.3 | 1.4 | 1.5 | 1.7 | 1.4 | 1.7 | 1.9 |
| WE | 88.9 | 116.9 | 130.7 | 118.8 | 172.0 | 223.7 | 1.3 | 1.5 | 1.8 | 1.4 | 1.9 | 2.1 |
| EE | 77.7 | 103.1 | 123.9 | 120.3 | 161.6 | 221.7 | 1.6 | 1.6 | 1.8 | 1.3 | 1.7 | 1.9 |
| SE | 84.2 | 111.9 | 129.3 | 115.7 | 169.3 | 231.8 | 1.4 | 1.5 | 1.8 | 1.4 | 1.8 | 2.1 |
| **Heatwave** | | | | | | | | | | | | |
| NE | 51.9 | 114.8 | 248.5 | 227.6 | 639.3 | 1639.5 | 4.3 | 4.5 | 5.6 | 2.2 | 2.7 | 3.0 |
| WE | 54.4 | 221.5 | 392.8 | 233.2 | 1269.6 | 2985.9 | 4.3 | 5.3 | 7.2 | 2.2 | 2.9 | 3.7 |
| EE | 65.9 | 207.1 | 385.8 | 299.4 | 1141.5 | 3066.8 | 4.5 | 5.4 | 7.9 | 2.2 | 2.9 | 3.6 |
| SE | 65.4 | 231.4 | 416.8 | 303.5 | 1493.8 | 4102.5 | 4.6 | 5.4 | 9.0 | 1.9 | 2.6 | 3.5 |
| **Compound** | | | | | | | | | | | | |
| NE | 4.9 | 15.2 | 30.0 | 10.4 | 44.1 | 128.7 | 2.1 | 2.8 | 4.5 | 0.1 | 0.2 | 0.5 |
| WE | 5.0 | 24.6 | 35.5 | 9.9 | 88.2 | 190.4 | 2.0 | 3.4 | 5.5 | 0.1 | 0.4 | 0.6 |
| EE | 6.6 | 21.4 | 34.9 | 15.0 | 72.5 | 188.6 | 2.3 | 3.4 | 5.5 | 0.1 | 0.3 | 0.6 |
| SE | 5.1 | 20.7 | 33.9 | 11.3 | 81.3 | 215.1 | 2.2 | 3.7 | 6.4 | 0.1 | 0.3 | 0.5 |
| **Cascading** | | | | | | | | | | | | |
| NE | 10.5 | 25.9 | 38.3 | 26.3 | 78.7 | 164.7 | 2.5 | 2.9 | 4.3 | 0.2 | 0.4 | 0.6 |
| WE | 10.7 | 35.4 | 40.3 | 25.5 | 123.9 | 212.4 | 2.4 | 3.5 | 5.4 | 0.2 | 0.6 | 0.7 |
| EE | 13.2 | 33.5 | 40.1 | 34.3 | 114.8 | 218.4 | 2.6 | 3.4 | 5.4 | 0.2 | 0.5 | 0.6 |
| SE | 10.7 | 31.1 | 38.6 | 26.9 | 119.9 | 242.2 | 2.5 | 3.7 | 6.3 | 0.2 | 0.5 | 0.6 |

**Figure 5.** A summary of drought, heatwave, and CnC characteristics in each European region for reference period, far future under SSP1-2.6, and far future under SSP5-8.5. The unit for drought duration is month, for heatwave is day, and for CnC is month. NE stands for northern Europe, WE stands for western Europe, EE stands for eastern Europe, and SE stands for southern Europe. Blue color indicates values under 25th percentile of each hazard characteristic from historical to SSP5-8.5 in NE, WE, EE, and SE; yellow color indicates median values of each hazard characteristic from historical to SSP5-8.5 in NE, WE, EE, and SE; red color indicates values above 75th percentile of each hazard characteristic from historical to SSP5-8.5 in NE, WE, EE, and SE.

to experience increase in both the number and duration of heatwave events under SSP5-8.5, with occurrences rising sixfold and durations extending thirteenfold compared to the reference period. Fischer and Schär (2010) similarly projected a high frequency of heatwave days in SE and EE, along with medium to high heatwave amplitudes in WE, especially in France. Projected average heatwave durations are anticipated to be 9 days in SE, 7.9 days in EE, 7.2 days in WE, and 5.6 days in NE. However, the highest frequency of heatwaves is predicted to occur in WE, with an average of 3.7 events per year, followed by EE with 3.6 events per year, SE with 3.5 events per year, and NE with 3 events per year. The increase of heatwave events and duration is lesser in NE compared to other regions, especially for SSP1-2.6 (bluish colors). On the contrary, the regional climate modeling study by Lin et al. (2022) suggests that EE will not experience the highest increase in heatwave magnitude; instead, NE is projected to have high increase. Nevertheless, their findings also support SE as a consistent hotspot region of future heatwaves.

Figure 5 also indicates that EE faced the highest number of CDH events from 1953 to 2014 (6.6 events), and an even greater number of CoDH events (13.2 events). Interestingly, the occurrences of CnC events under global warming are projected to be slightly higher in WE compared to EE (reddish colors under SSP1-2.6). Overall, the rate of increase in future CDH event will be nearly twice as high as the increase in CoDH events, though the absolute number of CoDH even remains greater. Future

CDH events are expected to be 4 to 6 times higher, while CoDH events are projected to rise by 3 and 3.5 times under SSP1-2.6 and SSP5-8.5 scenarios, respectively. Despite the higher number of future CnC events in WE and EE, these regions are not expected to have longer CnC durations. In contrast, SE is projected to experience the longest CnC durations across Europe under extreme warming conditions, with total durations of 215 months for CDH and 242 months for CoDH (Mukherjee and Mishra, 2021). The average duration of CnC events in SE is projected to be between 6.3 to 6.4 months. Additionally, the frequency of CnC events is expected to increase around 0.5 events per year in WE and EE for CDH events, and in WE for CoDH events. These findings are consistent with those of Mukherjee and Mishra (2021), who identified EE and Germany as hotspots for CDH frequency. Similarly, Tripathy et al. (2023) projected higher CDH events in Central Europe (stretched from western to eastern regions) followed by the Mediterranean and Northern Europe. This analysis highlights that while the number and frequency of future CnC events are higher in WE, SE is projected to experience longer durations of CnC events, consistent with previous study based on re-analysis data (Sutanto et al., 2019b).

### 3.4 Predicting drought and heatwave impacts in Germany

Predicting the impacts of dry hazards using Machine Learning (ML) approaches required an extensive dataset of documented impact occurrences (Mount et al., 2016). The impact of drought and heatwave events varies depending on their severity and duration, which means that not all events result in significant impacts. Furthermore, the availability of documented data on drought and heatwave impacts is often limited as in many countries impact data are not collected and reported at all, or only on an ad hoc basis. Germany stands out as the country with a sufficient number of documented drought impacts in the European Drought Impact Inventory (EDII) (Stahl et al., 2016; Sutanto et al., 2019a), which guided our selection of Germany as the focus for our impact prediction study.

Using the reported impacts in Germany and the Extreme Gradient Boosting (XGBoost) ML model (Friedman, 2001) trained using historical hazard indices (1953-2014), the models demonstrate robust forecasting performance, with area under the curve (AUC) values of 0.83, 0.76, and 0.81 for economic, non-economic, and ecosystem sectors, respectively (Supplementary Fig. S11a,b,c, see also Method). For heatwaves, the model evaluation shows a perfect score (AUC=1), which may be influenced by the limited amount of reported impact data (Supplementary Fig. S11d). We would like to note that the limited reported impact data can increase the overfitting risk because models can memorize the training data instead of learning generalizable patterns. This overfitting can lead to perfect discrimination (AUC=1) although it is not statistically robust. Simulations from drought impact models generally predict a higher number of impacts occurring within 1 month, as illustrated in Supplementary Figure S12a,b,c. However, these models tend to underestimate the duration of impacts for drought lasting longer than 5 months. For heatwaves, the model failed to predict impacts for the year 1987 and only predicted impacts with a maximum duration of 2 months (Supplementary Figure S12d). Overall, both drought and heatwave impact models provide relatively accurate simulations of past impacts, particularly for impacts occurred after the year 2000 due to more data on impacts.

Figure 6 illustrates simulations the number of drought and heatwave impacts under various scenarios for impacts occurring in a particular year. The number of drought impact on the economic sector is projected to increase in the future, rising from a median of 3 months to 4 months under both SSP1-2.6 and SSP 5-8.5, with longer impact up to 6 months (75th percentile)

projected for SSP5-8.5 (Fig. 6a). For non-economic sectors, the duration of drought impacts, which lasted between 2 and 4 months (25th and 75th percentiles, respectively) during the reference period, is projected to increase by 1 month under both SSP1-2.6 and SSP5-8.5 scenarios (Fig. 6b). The impact on the ecosystem sector is anticipated to increase from 3 to 4 months in the future under SSP5-8.5, based on the median ensemble (Fig. 6c). In contrast, the impact of heatwaves on humans, which rarely occurred annually in the past (Supplementary Fig. S12d) is predicted to become twice as long under SSP1-2.6 and four times under SSP5-8.5 (Fig. 6d). The projected increases in drought and heatwave impacts are likely driven by rising temperatures and less precipitation due to global warming, which can exacerbate heatwaves and accelerate soil drying, leading to increased soil moisture drought (Miralles et al., 2018; Teuling, 2018).

The characteristics of drought and heatwave impacts for both reference and future scenarios are detailed in Supplementary Table S3. The total number of drought impacts on the economic sector is projected to nearly double in the future, increasing from 36 events to 65 and 66 events under SSP1-2.6 and SSP5-8.5, respectively. The increase in the number of impacts corresponds to a longer total impact duration, extending from 128 months to 216 months for SSP1.2-6 and to 248 months for SSP5-8.5. Concerning average duration, however, SSP1-2.6 indicates a slightly shorter duration (0.2 months), while SSP5-8.5 demonstrates a longer duration by 0.3 months. The frequency (total number of events/total years) of drought impacts on the economic sector is expected to increase from 2.1 to 3.5 and 4 under SSP1-2.6 and SSP5-8.5, respectively. A similar trend is projected for the non-economic sectors, indicating more and longer impacts. The ecosystem sector is projected to be the most affected by droughts, with the number, total duration, and frequency of drought impacts being three times higher than the reference period. The impact of heatwaves on human health and losses, on the other hand, will sharply increase in the future due to projected higher temperatures (Peng et al., 2011; Amengual et al., 2014). For instance, the number of heatwave impacts is projected to increase tenfold under SSP5-8.5, with the total duration over 62 years being 25 times longer and the average duration three times longer (increasing from 1.1 to 2.9 months). The rise in the number and duration of impacts will also result in a higher annual frequency of events.

## 4 Discussion

Southern Europe is frequently identified as a hotspot for future drought and heatwave events (Forzieri et al., 2014; Prudhomme et al., 2014; Cammalleri et al., 2020). However, analyses on drought characteristics reveal that many regions across Europe, not just the south, will experience an increase in both the number and frequency of drought events in the future, particularly in Eastern European countries (Paparrizos et al., 2018). Our projections suggest that Southern Europe will experience longer drought periods, reinforcing its status as a drought hotspot (Sutanto et al., 2019b). Our findings on future heatwave characteristics also confirm that Southern Europe remains a hotspot for heatwaves (Amengual et al., 2014; Molina et al., 2020). Interestingly, while Southern Europe was not identified as a heatwave hotspot in the reference period, Eastern Europe and Russia were, likely due to the intense 2010 Russian heatwaves (Russo et al., 2015). In our study, we employed a relative metric based on 90th percentile of the historical climate, meaning that heatwaves could be identified everywhere in Europe even in the

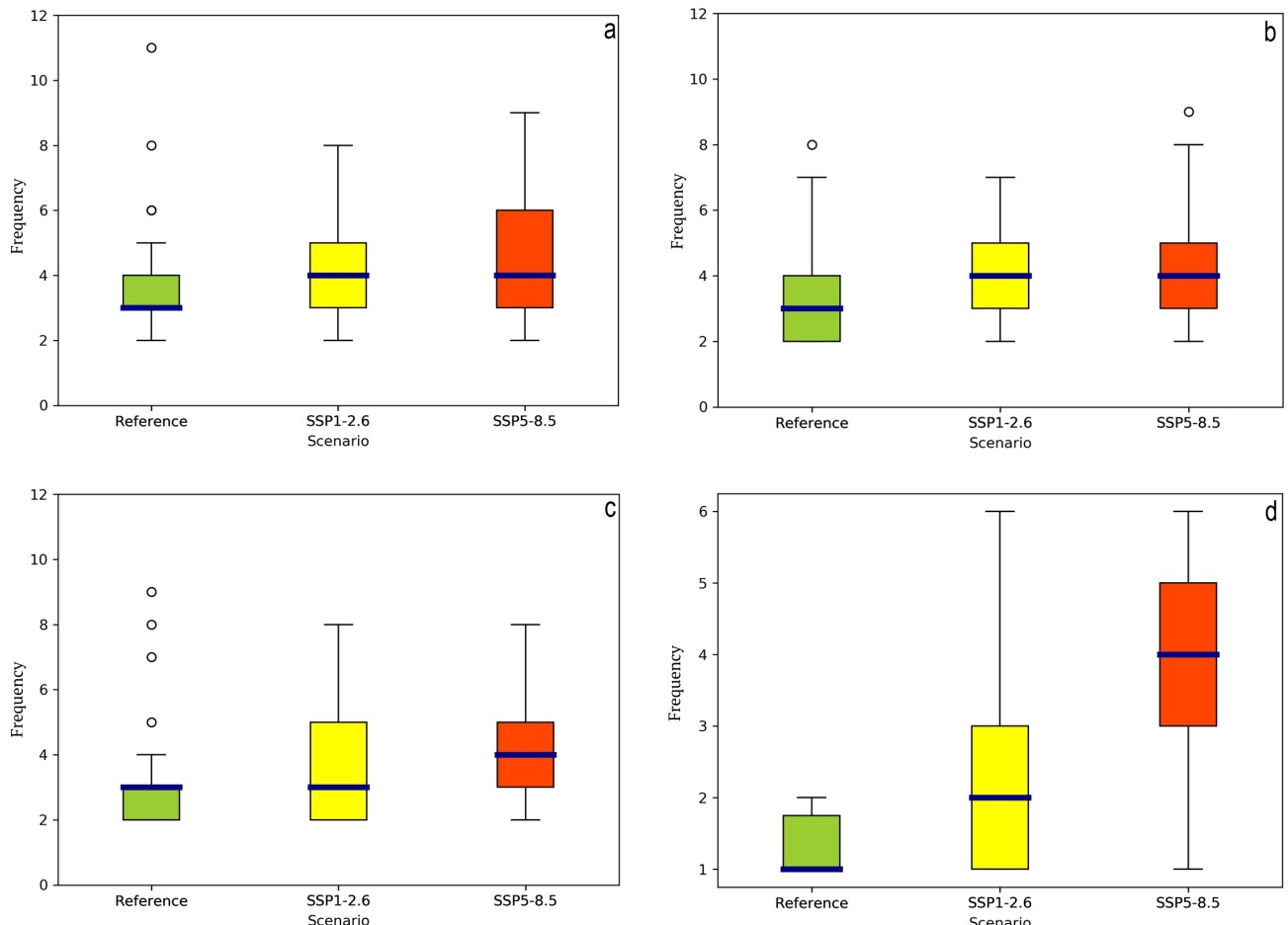

**Figure 6.** The predictions of drought and heatwave impacts in Germany under reference period (1953-2014) and future period (2039-2100) for SSP1-2.6 and SSP5-8.5. a) Prediction of drought impact on economic sector, b) Prediction of drought impact on non-economic sector, c) Prediction of drought impact on ecosystem sector, and d) Prediction of heatwave impact on humans. The Y axis shows the number of impacts in a year when impacts are predicted from all models. Lower box indicates 25 percentile, middle orange line indicates median, and upper box indicates 75 percentile. The whiskers show the 10 and 90 percentiles. Open circles indicate outliers.

cold climate regions. Changes in CnC characteristics follow similar patterns to drought characteristics, indicating that drought is the primary driver of CnC dry hazards (Sutanto et al., 2019b).

     While ML approach offers the advantage of directly supporting the development of hazard impact model and prediction, it is highly dependent on the availability of robust impact dataset. In this study, we acknowledge that the use of ML model to predict the occurrence of dry hazard impacts is constrained by the availability of impact data to train the model (Mount

et al., 2016). Despite combining impact data from the EDII (Stahl et al., 2016), the EM-DAT database (CRED, 2022), and

data mined from international journals and English reports, the compiled data remains limited, particularly for heatwaves. The database that was collected consisted of binary time series of impact occurrence (yes and no, 1 and 0). Thus, we could only predict the likelihood of impact occurrence, and no damage could be predicted. Additionally, all databases exhibit temporal and geographical biases (Bachmair et al., 2015; Stahl et al., 2016). Previous studies have highlighted missing data in many disaster databases, where either the disaster or its impact was not reported or were omitted due to low impact (Bachmair et al., 2016; Jones et al., 2023). The absence of impact reports does not always mean there were no impacts (Karlsson and Ziebarth, 2018). Consolidating drought impact data into three impact categories enhances the performance of drought impact model, particularly for predicting droughts lasting more than one month. Since impact models tend to overpredict impacts occurring only for one month duration, predictions of impacts of this duration should be excluded from the result. Moreover, longer drought events generally trigger more significant impacts than shorter ones. We identified only a few reported heatwave impacts, mainly related to human mortality, with very few reports on forest fires and agricultural losses. Overall, we obtained five years of reported heatwave impacts on human mortality, which could be predicted by the XGBoost model, except for the year 1987, resulting in an AUC curve close to 1. The small number of positive and negative samples allows the model to achieve an ideal measure of separability between these samples (Marcum, 1947).

The XGBoost ML technique offers several advantages over process-based models, such as increased accuracy and improved time and computational efficiency (Rahman et al., 2022; Peach et al., 2023). However, ML models to predict the impacts of CnC events could not be developed due to data constraints regarding both the number of event and their impacts. Droughts and heatwaves are extreme events, with a recurrence interval of approximately 10% for events with SMI<-1 and for the 90th percentile threshold, respectively (McKee et al., 1993; WMO, 2012). Furthermore, the occurrence of droughts and heatwaves, depending on duration and severity, does not always result in impacts (Bachmair et al., 2016), making the occurrence of drought and heatwave impacts even rarer (Seneviratne et al., 2021). Additionally, the occurrence of CnC events analyzed in this study is less than 10%. This scarcity of data is a primary reason why a CnC impact model could not be developed using the ML technique at the moment. Moreover, there is no existing database that provides information on CnC impacts. While EM-DAT includes information on primary and secondary hazards, it does not cover impacts. Thus, we highly recommend having a comprehensive multi-hazard impact database. Such a database would not only facilitate the development of ML-based impact models but also serve as a key component to validate the occurrence of natural hazards beyond reliance on empirical formulas alone.

Techniques for predicting drought impacts can be categorized into top-down, bottom-up, and hybrid approaches (Shyrokaya et al., 2023). The top-down approach utilizes a combination of climate models and deterministic impact models, such as employing crop models to predict yield losses (Ogutu et al., 2018; Sutanto et al., 2024). Sutanto et al. (2024) employed the WOFOST crop model to assess the impact of CnC events on maize yields, demonstrating the potential of this approach for impact prediction. The bottom-up approach relies on statistical models to link observed impacts to climate variables, such as using log regression model to predict drought impacts (Blauhut et al., 2015). The hybrid approach, employed in this study and others (Sutanto et al., 2019a, 2020b), is based on ML models to develop drought impact functions. The ML models can be utilized to predict various impacts and require less computational power than top-down approach, which is often dedicated to

only predict single impact. The hybrid approaches are valuable for developing impact-based prediction models, as predictions of impacts are still missing in many natural hazard early warning systems (EWSs) (Walker et al., 2024). Many EWSs provide information only on the occurrence of hazards and not on the corresponding impacts. Information on impact occurrence is of utmost importance for stakeholders and policymakers, providing actionable information for disaster risk management and risk reduction.

## 5    Conclusions and recommendations

This study contributes to an insight into predicting future drought and heatwave characteristics, both in isolation and as CnC events, including their impacts. Our findings indicate that drought and heatwave characteristics, such as the number, duration, and frequency of events, will increase across Europe under both SSP1-2.6 and SSP5-8.5. The increase in single hazard events will lead to a corresponding increase in CnC events. Historically, drought hotspots have been concentrated in West Europe, while heatwave hotspots have been identified in eastern and southern Europe, mainly in Russia, Italy, and Portugal. However, under future climate scenarios, these hotspots are projected to shift toward southern Europe. Our findings also demonstrate that the whole Europe will experience CnC events in the future, not just the southern regions where much of the focus of hot and dry hazard research has traditionally been concentrated. The occurrence of CDH events is expected to be six times higher than in the reference period, while CoDH events are projected to increase by 3.5 times under SSP5-8.5.

Additionally, this study highlights the potential of using a hybrid approach based on ML for projecting drought and heatwave impacts. We projected that drought impacts on economic, non-economic, and ecosystem sectors in Germany will be double in 2100, while heatwave impacts on human health and mortality will increase ninefold. These findings demonstrate that the hybrid method offers new opportunities not only for impact projection, as demonstrated here, but also for impact forecasting (Sutanto et al., 2019a; Shyrokaya et al., 2023). Although this study focuses on Europe, the approach is highly applicable to other regions increasingly vulnerable to CnC events, However, the success of such approach strongly depends on the availability of the completeness and high-quality impact data. We therefore advocate for the establishment of a standardized, global multi-hazard impact database to support improved ML model development for drought and heatwave impact-based forecasting.

To enhance Europe's resilience to future climate extremes, we recommend integrating CnC risk assessments into national and regional climate adaptation and disaster risk reduction strategies. This requires a shift from a hazard-centric approach toward multi-risk frameworks that account for CnC events and subsequently implementing effective adaptation strategies based on co-occurring events. The impact database, e.g., EDII, should be enhanced to provide information on the detailed reported damages, such as economic loses in Euro, number of people affected, yield reduction in T/ha, and percentage reduction in cargo ship. If this database exists, we believe that the ML models could be utilized to predict the damages. Regional planning should move beyond historically identified hotspots and address emerging risk zones, especially in southern and central Europe, where both hazard characteristics are projected to increase. Despite current limitations in hybrid approach for predicting CnC impacts, advancing complementary approaches—including top-down, bottom-up, and hybrid methods—will be essential for building a comprehensive and actionable understanding of future multi hazard risks.

*Code availability.* All codes used to conduct the analysis presented in this paper will be available online in the 4TU Centre for Research

Data with https://doi.org/xxxx (Sutanto and Duku, 202x).

*Data availability.* The ERA5 Land data are accessible through the Copernicus Data Store (CDS), which are freely available (doi:10.24381/cds.e2161bac). Soil moisture and projection data are downloaded from the ISIMIP output (ISIMIP3b) for global water sector (https://data.isimip.org/). Other data and generated and/or analyzed during this study will be available online in the 4TU Centre for Research Data with https://doi.org/xxxx (Sutanto and Duku, 202x).

*Author contributions.* All authors conceived and implemented the research. Data analyses, model output analyses, and all figures have been performed by S.J.S. and M.G. The ML models were developed by C.D. S.J.S wrote the initial version of the paper. R.D. and S.P contributed to interpreting the results, discussion, and improving the paper.

*Competing interests.* The authors declare no competing interests.

*Acknowledgements.* The research is supported by the CDHEU project, which is funded by the Wageningen Data Driven Discoveries in

Changing Climate (D3-C2). The hydrometeorological outputs came from the Copernicus Data Store and the ISIMIP data repository (see data availability section). All data were processed using High Performance Computing Cluster Anunna hosted by Wageningen University and Research.

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
