# Peer review of "Future intensification of compound and consecutive drought and heatwave risks in Europe"

_EGUsphere, 2025_

## Author Response (AR1)

**Reply to reviewers**

We would like to thank the reviewers for valuable suggestions and comments. We reply to each of these with page number and line indications based on the clean manuscript (no track changes). **P** refers to the page number and **L** refers to the line number. For example, **P3L65-70**, refers to page 3, lines 65-70.

| Reviewer 1 | | |
|---|---|---|
| **No** | **Comment** | **Reply** |
| 1 | This article is in line with the current research on climate risks, which builds on the single hazard perspective to work with a multi-hazard perspective. This work is very interesting and a very valuable contribution to the current research on climate risks. It builds on theme of multi-hazards, expanding the knowledge on compound and cascading drought and heatwave risks, with also a novelty in the approach, using machine learning. The article is well written and easy to read. The structure of the article is also coherent and easy to follow. | We would like to thank the referee for the acknowledgement of the novelty of our paper, contributing to the current research on climate risk and multi hazard framework. |
| 2 | Introduction: from my perspective the introduction is interesting and includes many good references but lacks references from a "higher-level perspective". I would suggest to includes references to the most recent reports of the IPCC, IPBES or the European climate risks assessment report (European Climate Risk Assessment \| European Environment Agency's home page). | In our manuscript, we cited IPCC AR6 report, which is the latest IPCC report. However, we would like to acknowledge the authors of working group 1 where the text was cited. Thus, we cited the IPCC report as Seneviratne et al. (2021) instead of IPCC (2021). EEA (2024) and IPBES (2021) references have been added in the revised version (**P2L26**). |
| 3 | More information about machine-learning: I suggest including more discussion on the advantages and drawbacks of using machine-learning versus other methodologies/tools. This discussion could be part of section 4 or 5 but it could also be part of the Introduction, for example, in the third paragraph. What tools were used before to do this kind of estimate? Why do we need ML? What are the advantages? What are the drawbacks? Can we validate the results? How? It is already partly covered in the article but I think it would be further discussed. | We thank the reviewer for his/her valuable suggestions. We expanded the introduction section, explaining the machine learning approach used in Stagge et al. (2015), Bachmari et al. (2017), and Sutanto et al. (2019a) (**P2L54-P3L61**). We discussed the advantages and disadvantages of ML approach, such as offering direct impact prediction and required robust impact data in the discussion section, paragraph 2 and 3 (**P16L358-378**). |
| 4 | In Section 2: The definition of cascading events is not very clear to me. This part of the section needs more explanation, maybe not a detailed description like in the previous articles you mentioned, but a more detailed description is important as this is a central part of the article. | The definition of compound and cascading events employed in this study was expanded in the Section 2.4 (**P5L137-150**). In addition, Table 1 showing examples of compound and cascading events was added so readers can understand the definition easily (**P6**). |
| 5 | Section 3.3: When you summarize the results in the table, it would also be good to say how these results compare to the results from existing literature and | Suggestion is accepted. We compared our findings with previous literature on drought, heatwave, and compound projection literature. In general, our findings align with studies from Samaniego et al. (2018) who |

| | | |
|---|---|---|
| | potentially comment if there are differences. | found that EE and SE will experience higher and longer droughts (**P12L264-265**). Similarly, our heatwave findings are in agreement with a study conducted by Fischer and Schär (2010) and Lin et al. (2022) for SE, EE, and WE. However, we did not project that NE will experience high heatwave as it is found in Lin et al. (2022) (**P13L277-285**). For CDH results, we confirmed the findings with literature conducted by Mekherjee and Mishra (2021) and Tripathy et al. (2023) (**P13L296-298**). |
| 6 | Description of the scenario: I might have missed it, but I don't think I found a description of the scenarios RCP-SSP. It does not need to be long, but it might be good to briefly describe what these scenarios mean. It could be in section 2 for example. | The reviewer is correct. We overlooked the climate scenarios since we assumed the readers are familiar with this. We added information of SSP scenarios in Section 2.1 (**P3L82-83**). |
| 7 | Line 19: I think a word might be missing here, when you say "economic, non-economic and ecosystem". Economic impacts? Sectors? I would recommend reformulating this sentence. | We thank the reviewer for careful reading. We agree that the sentence misses the word sectors. We revised the sentence accordingly (**P1L20**). |
| 8 | Line 22: "urgency of climate mitigation". Climate adaptation could also be mentioned here. | We revised the sentence into: "…urgency of climate adaptation and mitigation…" (**P1L22**). |
| 9 | The concluding section is quite short, I wish it would include a few recommendations for the future. What data do you need to make your work adaptable to Europe in general? Outside Europe? It is complementary to other technics/tools/methods? Can this information be used by decision makers? If yes, how? In which context? | The conclusion is short because we would like to make it concise by only describing the main findings of our study. However, we agree, and we added one paragraph about recommendation. Thus section 5 became conclusion and recommendation section.

Information regarding the applicability of our approach to outside Europe was added. We also suggest for the establishment of a standardized, global multi-hazard impact database to support improved ML model development for drought and heatwave impact-based forecasting. We also recommend to integrate CnC risk assessments into national and regional climate adaptation and disaster risk reduction strategies. Furthermore, we suggest that regional planning should move beyond historically identified hotspots and address emerging risk zones, especially in southern and central Europe, where both hazard characteristics are projected to increase (**P18L416-423**). |
| 10 | Figure
• Modifying the colormap could be good, there might be other colormaps to use where it is easier to see the positive vs negative differences. | We modified the colormap in the revised version to improve its readable. Moreover, the sub-titles have been added. The plotting boundaries, right and bottom, have been cut, thus removing the white areas outside the study regions. |

| | |
|---|---|
| • I would suggest including sub-titles for the different panels when you have several figures so the reader doesn't have to look at the legend every time.

Is it possible to fit the figures where you have results? There are white areas on the right sides of figures and down which are usually removed. | |

**References**

Seneviratne, S. I., Zhang, X., Adnan, M., Badi, W., Dereczynski, C., and co authors: Weather and Climate Extreme Events in a Changing Climate. In Climate Change 2021: The Physical Science Basis. Contribution of Working Group I to the Sixth Assessment Report of the Intergovernmental Panel on Climate Change [Masson-Delmotte, V., P. Zhai, A. Pirani, S.L. Connors, C. Péan, S. Berger, N. Caud, Y. Chen, L. Goldfarb, M.I. Gomis, M. Huang, K. Leitzell, E. Lonnoy, J.B.R. Matthews, T.K. Maycock, T. Waterfield, O. Yelekçi, R. Yu, and B. Zhou (eds.)], Cambridge University Press, Cambridge, United Kingdom and New York, NY, USA, pp. 1513–1766, https://doi.org/https://doi.org/10.1017/9781009157896.013, 2021.

| Reviewer 2 | | |
|---|---|---|
| **No** | **Comment** | **Reply** |
| 1 | This constitutes an interesting study, however I do have some concerns regarding the downscaling procedures applied. Further, the authors should look spend more time in investigating the model skill of used ISIMIP data compared to ERA5. Additional comments relate to the impact projections which are provided in units of time not damage. The paper should also be proofread again, to remove a few remaining grammatical errors. | We thank the referee for his/her positive interest in our study, and support in improving our manuscript.

Regarding the downscaling, we employed the bilinear interpolation approach on the ISIMIP datasets. We did not apply statistical or dynamical downscaling techniques, and as such the resampling of the ISIMIP data did not substantially change the climate change signal that is contained in these data (**P3L84-P4L90**). To avoid any further confusion, we renamed the downscaling into resampling.

Our study does not aim to evaluate the performance of ISIMIP models compared to ERA5. However, we utilized ERA5 Land soil moisture data for bias corrected the soil moisture data simulated by CWAT model forced with ISIMIP climate models (**P3L83-84**). This approach is commonly used in many studies dealing with climate change datasets.

Employing the machine learning approach to predict drought and heatwave impacts will result, in general, likelihood of impact occurrences (LIO) as presented by previous studies (e.g., Stagge et al., 2015; Blauhut et al., 2015; Bachmair et al., 2017; Sutanto et al., 2019a). The machine learning approach utilized in this study only uses binary time series of impact occurrences, yes or no impact (**P2L54-P3L59**). Furthermore, we combined impact data from different sectors due to data limitation. By doing this, no damage can be predicted. We suggest that impact database should provide detailed reported damage. If the damage data becomes available, future study could utilize this dataset for damage predictions (**P18L423-425**). We added this information in the introduction and recommendation sections in the revised version. |
| 2 | l.3-4 Not clear what this sentence is trying to say | We have rewritten the sentence into: "Yet, most studies on drought and heatwave have focused on single hazard events rather than compound and cascading events and their potential impacts" (**P1L3-4**). |
| 3 | l.14 wrong grammar 'in the west europe' | It is now written as "in western Europe" (**P1L14**). |
| 4 | l.35 it should be defined what the authors consider compound and cascading hazards. Are these temporally concurrent events or sequential events or both? The term 'cascading' implies a causal relationship between both events (i.e. a trigger – | We thank the reviewer for the suggestion. The definition of compound and cascading events was mentioned in the first paragraph. We define compound event if drought and heatwave occurred at the same time and place (concurrent) and cascading event if |

| | | |
|---|---|---|
| | response dynamic) and should not be used if event relationships are investigated stochastically, only. | drought and heatwave occurred one after another at the same time and place (sequential) (**P2L26-28**). Furthermore, we explained the definitions in a more detailed manner in the section 2.4. These definitions have been applied in previous studies (Leonard et al., 2014; Liu and Huang, 2015; Vitolo et al., 2019; Sutanto et al., 2019) (**P5L137-141**). |
| 5 | l.55 agricultural droughts and hydrological droughts are different things, I would suggest to just use 'drought' here defined by soil moisture deficiency. | We used the term hydrological drought and removed the word agriculture (**P3L62-63**). Soil moisture is one of the hydrological components and therefore, we prefer to identify soil moisture drought as hydrological drought instead of agricultural drought. |
| 6 | l.58 – 61 as mentioned above I would suggest to stick with the compound event typology described in Zscheischler et al. 2021 and elsewhere by refereeing to these two event types as temporally compounding (consecutive events over same place) and spatially compounding event (concurrent events over same place). | We understand that some studies used the term compound event only to indicate both the events that are concurrent and simultaneous. However, we prefer to split this definition into two: compound and cascading. If drought occurs after heatwave event is over (here the temperature back to normal-high, not extreme), then we define this event as cascading and not compound/concurrent because there is only one single hazard left in the end (see point 4). We further clarified this definition in the method section (**P5L137-141**). Reference Zscheischler et al. (2020) was added. |
| 7 | l.77 downscaling the low. Res. data (drought) instead of upscaling the high. Res. data, gives a wrong sense of accuracy. Results should be investigated at the lowest resolution available. | We thank the reviewer for the feedback. The rationale behind the downscaling (will be resampling) soil moisture and temperature data is to achieve high resolution results, which is needed for sectoral applications. Figure 1 below shows the difference between results using ISIMIP resolution (100 km) and ERA5 Land resolution (10 km). It is obvious that high resolution data will have better impression for discussing about natural hazard impacts with stakeholders. Moreover, we aim to use drought and heatwave indices to develop impact prediction algorithms using machine learning and impact data at the national level. Using a coarse resolution for impact prediction will result in limited number of grid cells. We described this in the method section (**P3L86-88**). |
| 8 | l.99 sentence seems wrong: 'wrong data mined (?).. ' | We revised the word to "data mining" (**P4L112**). |
| 9 | l.124. split sentence, it is hard to understand. | We split the sentence into "To analyse the CnC events, binary maps consist of the number 1 for heatwave and 2 for drought were generated if the month is identified as drought or heatwave month. For no hazard month, 0 value is applied" (**P5L142-143**). |
| 10 | Figures 1-4 how do ISIMIP models perform against ERA5 for drought / heatwaves and compound events over the historical | In this study, we did not evaluate the performance of ISIMIP models for identifying drought and heatwave characteristics compared to ERA5. The goal of our study is to |

| | | |
|---|---|---|
| | period? This deserves a paragraph and at least some figures in the SI. | analyze the changes in drought and heatwave characteristics including their compounding events in a warming world. Some previous studies also utilized the ERA5 datasets for downscaling and bias corrected ISIMIP model. We suggest that future study may focus on the performance of ISIMIP models in identifying drought and heatwave compared to ERA5. |
| 11 | Figures: please don't use rainbow color scales, see link for reasons: https://blogs.egu.eu/divisions/gd/2017/08/23/the-rainbow-colour-map/ | We revised the colormap as it is also suggested by reviewer 1. |
| 12 | Table / Figure 5: Where are these Regions? This should be marked in the Figures 1-4 or an additional figure with defined regions should be provided in the SI. | The regions are presented in the Supplementary Figure S9. We mentioned this in **P11L260-261**. |
| 13 | l.257 'What is more' is not a usual expression. | I think the reviewer means L275. We revised the word into "furthermore" (**P13L304**). |
| 14 | l.283 "For heatwaves, the model evaluation shows a perfect score (AUC=1), which may be influenced by the limited amount of reported impact data (Supplementary Fig. S10d)." This seems odd, how can a small sample size lead to a perfect model performance? Please explain. | We thank for the valuable feedback. The AUC can generate a value 1 when the sample size is small. First, the AUC measures the ability of a classifier to rank a randomly chosen positive instance higher than a randomly chosen negative one. If we have 2 positive and 2 negative samples and the model predicts these correctly by "accident" then the AUC will be 1 although it is not statistically robust. Second, with a small sample, there is an overfitting risk. With very small datasets, models can memorize the training data instead of learning generalizable patterns. This overfitting can lead to perfect discrimination. We explained this issue in the revised version (**P14L313-315**). |
| 15 | Results in Fig. 6 I don't understand why impacts are provided in units of time. Impacts should be measured as monetary damage e.g. in currency (econ. Impacts), or excess mortality (health impacts). The y-axis units in Figure six are not provided and 'Number of Impact' is probably Grammarly wrong. | As explained in point 1, the machine learning approach utilized in this study only uses binary time series of impact occurrences, yes (1) or no impact (0). The reported impact database such as EDII does not provide detailed economic damage per sector so we could not predict the damage. If the damage data becomes available, future study could utilize this dataset for damage predictions. The Y axis shows the occurrence of impact in a year when impacts are predicted from all models. |
| 16 | l.377 this is an overstatement. There are numerous studies on compound drought and heat occurrences, which should be cited here. A simple search in google scholar will reveal numerous papers. | We are not sure, which sentence that the reviewer referring to. L377 is "We projected that drought impacts on economic, non-economic, and ecosystem sectors in Germany will be double in 2100, while heatwave impacts on human health and mortality will increase ninefold." In this sentence, we refer to drought and heatwave impacts and not events. In addition, previous studies on drought and heatwave events in Europe |

| | | support our findings that both events will increase due to climate change. |
|---|---|---|

---

## Author Response (AR2)

**Reply to reviewer 2**

We would like to thank the reviewer for valuable suggestions and comments. After the paper has been revised, we will reply to each of these with page number and line indications. **P** refers to the page number and **L** refers to the line number. For example, **P3L65-70**, refers to page 3, lines 65-70.

| Reviewer 1 | | |
|---|---|---|
| **No** | **Comment** | **Reply** |
| 1 | I thank the authors for their response, which have addressed some of my minor concerns. However, my main concerns were not adequately addressed. I therefore restate these comments below and hope that the reviewers will find the time to address these in a future revision. | We thank the reviewer for the comments. Although we do not fully agree with all the points raised, we appreciate the opportunity to reflect on them. In our responses below, we aim to provide constructive clarifications and propose intermediate solutions that address the reviewer's concerns while preserving the core objectives of the manuscript. |
| 2 | My concerns about the usability of these results persist, as the authors do not show evidence that ISIMIP is providing reliable results. Benchmarking the used climate model data against observations is not difficult and but very important in the context to discuss the reliability of the shown results. (see my comment 10) | We acknowledge the concern regarding the reliability of ISIMIP outputs compared to observations. However, we would like to clarify that the primary objective of this study is not to evaluate ISIMIP models, but to use them as input data for assessing the impacts of droughts and heatwaves. This is the main reason why we did not include a dedicated ISIMIP model validation in our manuscript.

Nonetheless, we agree that we should provide information regarding the ISIMIP model evaluations.

The ISIMIP models have been extensively used in recent years to study the impacts of climate change on the hydrological cycle and water resources (E.g., Eisner et al., 2017; Vetter et al., 2017; Mishra et al., 2017; Wang et al., 2017; Gelfan et al., 2017). More specifically, it has been applied to study the extreme events, such as droughts, floods, and heatwaves (Samaniego et al., 2017; Pechlivanidis et al., 2017; Tabari et al., 2021; Messori et al., 2025), suggesting its robustness for such applications.

Several studies have investigated uncertainties within the ISIMIP models and have concluded that most of the variability stems from the climate models rather than the hydrological models (Samaniego et al., 2017; Vetter et al., 2017; Hattermann et al., 2018). Furthermore, uncertainties tend to be higher in dry basins than in wet basins (Samaniego et al., 2017; Pechlivanidis et al., 2017). We added this information in the revised version. |

| | | | We also included a comparison between drought and heatwave analysis derived from ISIMIP models and ERA5 Land. The results show that the simulated number of drought events from ISIMIP models aligns closely with ERA5 Land, with a median difference of only 7% (Figure 1a below). For heatwaves, ISIMIP models slightly underestimate their frequency compared to ERA5 Land, with 75 percentile of events reaching 75 in ERA5 land and 59 in ISIMIP models (Figure 1b). These findings support previous studies, which report higher uncertainty from the climate models than the hydrological model. In addition, the bias corrected ISIMIP datasets used in our study show lower uncertainty compared to ERA5 Land (see also point 3). We incorporated this explanation into the revised method section and added Figure 1 to the appendix (**P4L98-108**). |
|---|---|---|---|
| 3 | | I'm also not satisfied with the authors response to my question regarding the increasing the resolution by a regridding the data. To be convinced that this is useful approach, I would like to see the differences between results based on the original low-res grid and the artificially increased high res. results. Of course stakeholders are interested in high res data, but if this data consists of artefacts then sticking with more reliable low res data is the way to go (original comment 7). | We regret that our previous response regarding the regridding of data did not fully satisfy the reviewer and we will clarify in more detail below.

The main reason we chose to conduct our analysis using high resolution data is directly related to the bias correction process applied. The regridding was done to facilitate an easy bias-correction of the ISIMIP soil moisture data with ERA5-Land data, and should not be regarded as an attempt to downscale the ISIMIP data. The ISIMIP soil moisture data were first resampled to match the ERA5-Land resolution, and subsequently bias-corrected using ERA5-Land soil moisture data as the observational reference (**P3L84-87**). This step was essential to ensure that the model outputs better reflect real-world conditions.

It is important to note that the uncorrected ISIMIP soil moisture values were lower than ERA5 Land, largely due to differences in model structures, soil layering, and depth representations (see Figure 2 below). After bias correction, the modeled soil moisture data show a much closer agreement with ERA5 Land data.

A comparison using the original low resolution data would not yield a meaningful evaluation, since only the high resolution data underwent bias correction. Given these points, we believe that working with bias corrected high resolution data was more appropriate and scientifically sound |

| | | approach for our analysis. We are confident that these data better represent the observed soil moisture conditions that the original ISIMIP outputs and thus are more suitable for the objectives of our study. |
|---|---|---|
| 4 | I can only repeat my initial comment that the term 'cascading' should not be used for 'sequential' events. In their response to my initial comment 4 and 6 the author use sequential to explain what they mean with cascading. I would therefore urge the authors to change 'cascading' to 'sequential' throughout the manuscript. | The feedback from the reviewer is accepted. We thus revised the word cascading into consecutive throughout the manuscript. |
| 5 | Figure 6 then does not show an impact but a frequency (my original comment 15). Please change the Figure label accordingly. | We changed the figure Y axis accordingly. |
| 6 | Original comment 16, my apologies for the mix up I meant l. 367 'This study is pioneering in predicting future drought and heatwave characteristics, both in isolation and as CnC events, including their impacts.' Compound Drought heat events are one of the most thoroughly studied compound event types. While I do not want to downplay the ambitions of the researchers 'pioneering' seems a bit much here. | We changed the sentence into "This study contributes to an insight into predicting future drought and heatwave characteristics, both in isolation and as CnC events, including their impacts" (**P18L417-418**). |

**References**

Eisner, S., Flörke, M., Chamorro, A., Daggupati, P., Donnelly, C., Huang, J., Hundecha, Y., Koch, H., Kalugin, A., Krylenko, I., Mishra, V., Piniewski, M., Samaniego, L., Seidou, O., Wallner, M., and Krysanova, V.: An ensemble analysis of climate change impacts on streamflow seasonality across 11 large river basins, Clim. Change, 141, 401–417, https://doi.org/10.1007/s10584-016-1844-5, 2017.

Gelfan, A., Gustafsson, D., Motovilov, Y., Arheimer, B., Kalugin, A., Krylenko, I., and Lavrenov, A.: Climate change impact on the water regime of two great Arctic rivers: modeling and uncertainty issues, Clim. Change, 141, 499–515, https://doi.org/10.1007/s10584-016-1710-5, 2017.

Hattermann, F. F., Vetter, T., Breuer, L., Su, B., Daggupati, P., Donnelly, C., Fekete, B., Flörke, F., Gosling, S. N., Hoffmann, P., Liersch, S., Masaki, Y., Motovilov, Y., Müller, C., Samaniego, L., Stacke, T., Wada, Y., Yang, T., and Krysnaova, V.: Sources of uncertainty in hydrological climate impact assessment: a cross-scale study, Environ. Res. Lett., 13, 015006, https://doi.org/10.1088/1748-9326/aa9938, 2018.

Messori, G., Muheki, D., Batibeniz, F., Bevacqua, E., Suarez-Gutierrez, L., and Thiery, W.: Global mapping of concurrent hazards and impacts associated with climate extremes under climate change, Earth's Future, 13, e2025EF006325, https://doi.org/10.1029/2025EF006325, 2025.

Mishra, V., Kumar, R., Shah, H. L., Samaniego, L., Eisner, S., and Yang, T.: Multimodel assessment of sensitivity and uncertainty of evapotranspiration and a proxy for available

water resources under climate change, Clim. Change, 141, 451–465, https://doi.org/10.1007/s10584-016-1886-8, 2017.

Pechlivanidis, I. G., Arheimer, B., Donnelly, C., Hundecha, Y., Huang, S., Aich, V., Samaniego, L., Eisner, S., and Shi, P.: Analysis of hydrological extremes at different hydro-climatic regimes under present and future conditions, Clim. Change, 141, 467–481, https://doi.org/10.1007/s10584-016-1723-0, 2017.

Samaniego, L., Kumar, R., Breuer, L., Chamorro, A., Flörke, M., Pechlivanidis, I. G., Schäfer, D., Shah, H., Vetter, T., Wortmann, M., and Zeng, X.: Propagation of forcing and model uncertainties on to hydrological drought characteristics in a multi-model century-long experiment in large river basins, Clim. Change, 141, 435–449, https://doi.org/10.1007/s10584-016-1778-y, 2017.

Tabari, H., Hosseinzadehtalaei, P., Thiery, W., and Willems, P.: Amplified drought and flood risk under future socioeconomic and climatic change. Earth's Future, 9, e2021EF002295, https://doi.org/10.1029/2021EF002295, 2021.

Vetter, T., Reinhardt, J., Flörke, M., van Griensven, A., Hattermann, F., Huang, S., Koch, H., Pechlivanidis, I. G., Plötner, S., Seidou, O., Su, B., Vervoort, R. W., and Krysanova, V.: Evaluation of sources of uncertainty in projected hydrological changes under climate change in 12 large-scale river basins, Clim. Change, 141, 419–433, https://doi.org/10.1007/s10584-016-1794-y, 2017.

Wang, X., Yang, T., Wortmann, M., Shi, P., Hattermann, F., Lobanova, A., and Aich, V.: Analysis of multi-dimensional hydrological alterations under climate change for four major river basins in different climate zones, Clim. Change, 141, 483–498, https://doi.org/10.1007/s10584-016-1843-6, 2017.

[Figure]

**Figure 1**. a) Comparison of total number of drought events between ERA5 Land and ISIMIP models across Europe from 1953 to 2014 and b) Comparison of total number of heatwave events between ERA5 Land and ISIMIP models across Europe from 1953 to 2022.

[Figure]

**Figure 2**. Comparison ISIMIP modeled soil moisture data with ERA5 Land without (a) and with bias correction (b).